# Reliable likelihoods from conditional flow matching generative models in feature space

## Abstract

Normalising flows are a flexible class of generative models that provide exact likelihoods, and are often trained through maximum likelihood estimation. Recent work suggests that discrete-step flow models trained in this way can assign undesirably high likelihood to out-of-distribution image data, bringing their reliability for applications where likelihoods are important (e.g. outlier detection) into question. We show that continuous-time normalising flows trained with the conditional flow matching objective (CFM models) also provide un-reliable likelihoods, and then investigate whether CFM models trained on various feature representations can lead to more reliable likelihoods. We consider (1) the original data; (2) features from a pretrained classifier; (3) features from a pretrained perceptual autoen-coder; and (4) features from an autoencoder trained with a simple pixel-based reconstruction loss. Our proposed pixel autoencoder representations lead to reliable likelihoods from CFM models on out-of-distribution data but can yield samples of lower quality, suggesting opportunities for future work.

## 1 Introduction

Normalising flows are generative models that specify a target density through a base distribution and an invertible transformation process, with applications in computer vision (Kingma & Dhariwal, 2018; Dinh et al., 2017; Lipman et al., 2023; Kumar et al., 2020; Müller et al., 2019; Abdelhamed et al., 2019), audio generation (Esling et al., 2019; Kim et al., 2018; Prenger et al., 2019), graph generation (Madhawa et al., 2019), reinforcement learning (Mazoure et al., 2020; Ward et al., 2019; Touati et al., 2019) and physics (Kan-war et al., 2020; Köhler et al., 2019; Noé et al., 2019; Wirnsberger et al., 2020; Wong et al., 2020). They offer exact likelihood evaluation as an advantage over other generative models, enabling, in principle, outlier detection. Of interest to us is the peculiar phenomenon of normalising flows assigning undesirably high likelihoods to out-of-distribution data (Nalisnick et al., 2019a; Kirichenko et al., 2020; Voleti et al., 2024), which brings their reliability for applications into question.

A discrete-step normalising flow specifies a target distribution $p_x(\boldsymbol{x})$ in terms of an easy-to-sample-from base distribution $p_u(\boldsymbol{u})$, and an invertible transformation $\boldsymbol{u} = g(\boldsymbol{x})$ with $\boldsymbol{u} \sim p_u(\boldsymbol{u})$, by employing the change-of-variables formula. $g(\boldsymbol{x})$ is defined as a composite function, usually a neural network whose architecture is restricted for a tractable log-determinant in the change-of-variables formula. The continuous-time variant (Chen et al., 2018; Grathwohl et al., 2019), hereafter referred to as a continuous flow, expresses $\boldsymbol{u} = g(\boldsymbol{x})$ as the solution to an initial value problem (IVP):

$$\frac{d\boldsymbol{z}}{dt} = f_{\boldsymbol{\theta}}(\boldsymbol{z}, t), \qquad t \in [t_0, t_1], \qquad \boldsymbol{z}_0 = \boldsymbol{z}(t_0) = \boldsymbol{u} \qquad (1)$$

with the data represented by the solution at time $t_1$, i.e. $\boldsymbol{z}_1 = \boldsymbol{z}(t_1) = \boldsymbol{x}$. A continuous analog of the change-of-variables formula is used to determine $\log p_x(\boldsymbol{z}_1)$ (Chen et al., 2018). The function $f_{\boldsymbol{\theta}}(\boldsymbol{z}, t)$ defines a time-dependent vector field describing the transformation dynamics, with trainable parameters $\boldsymbol{\theta}$. This formulation circumvents restrictions on $g(\boldsymbol{x})$ for a tractable log-determinant, at the time-cost of simulating solution trajectories for the IVP in Equation 1.

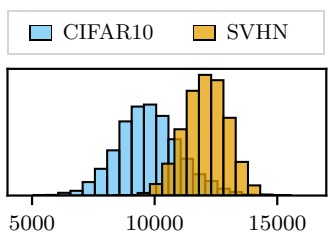 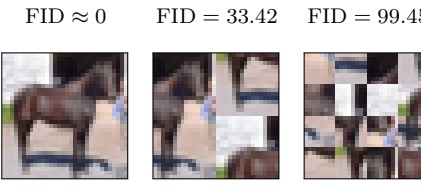 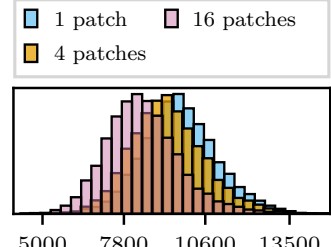

(a) log likelihoods from a CFM model trained on CIFAR10

(b) random patch shuffles of in-distribution test images lead to the histograms of log likelihoods shown on the right

Figure 1: The normalised log likelihood histograms in (a) indicate that a CFM model trained on CIFAR10 assigns higher likelihood to out-of-distribution data from SVHN, compared to in-distribution CIFAR10 test data. The overlapping histograms in (b) indicate that the same model assigns similar likelihoods to various levels of patch-shuffled CIFAR10 test images, despite FID scores that suggest changes in semantic content.

Continuous flows trained with the recently introduced conditional flow matching (Lipman et al., 2023; Tong et al., 2024) objective (CFM models) circumvent maximum likelihood training and the need for simulating solution trajectories. With the bottleneck of simulation removed, continuous flows become more relevant to applications at scale. But it turns out that CFM models also assign unreliable likelihoods to out-of-distribution data, as we demonstrate in Figure 1a, where a model trained on the CIFAR10 dataset assigns higher likelihoods to samples from the SVHN dataset (the histogram corresponding to SVHN is mostly to the right of the one corresponding to CIFAR10, indicating larger log likelihood values). The continuous nature of the invertible transformations introduces distinct inductive biases compared to discrete-step normalising flows, prompting further investigation into the reliability of likelihood estimates in continuous flows. Motivated by previous observations that unreliable likelihoods may stem from certain structures in representations of the data (Ren et al., 2019; Serrà et al., 2020; Kirichenko et al., 2020; Zhang et al., 2023), we will explore whether learned representations of the input data can lead to more reliable likelihoods.

The tendency of CFM models to produce unreliable likelihoods, specifically when trained on pixel representations of images, can be further demonstrated by applying an experiment of Voleti et al. (2024) to these models. In Figure 1b we show normalised log likelihood histograms obtained from a CFM model trained on CIFAR10, for patch-shuffled versions of the CIFAR10 test set. The Fréchet inception distance (Heusel et al., 2017) of these datasets increases with the number of shuffled patches, indicating a change in semantic content, yet model likelihoods are affected to a much lesser degree.

For discrete-step normalising flows, Kirichenko et al. (2020) managed to improve the reliability of likelihoods by training on feature representations from a classifier pretrained on ImageNet. Such feature representations are convenient but not without limitation. Firstly, the pretrained network may struggle to generalise to data that differs in distribution from ImageNet. Secondly, while a flow model trained in feature space can be used to generate new feature vectors, the absence of a decoder makes them unsuitable for image generation. To address these limitations, we will train and evaluate CFM models on a number of representations: (1) EfficientNet-B4 features, similar to what Kirichenko et al. (2020) did for discrete-step flows; (2) features obtained from a pretrained perceptual autoencoder (Rombach et al., 2022); and (3) features obtained from an autoencoder trained from scratch with a simple pixel-based loss. Autoencoders provide an ability to decode from feature space back to image space, thereby enabling the generation of image data. Our findings can be summarised as follows:

1. Training CFM models on EfficientNet-B4 features results in slightly more reliable likelihood estimates for out-of-distribution data. However, this approach does not fully resolve the issue and restricts the ability to generate images.

2. CFM models trained on feature representations from the pretrained perceptual autoencoder can generate (qualitatively) good samples, but do not solve the likelihood reliability problem altogether. We also note some perceptual similarity between these feature representations and the corresponding input images.

3. In contrast, feature representations from the pixel autoencoder lead to a consistent improvement in the reliability of likelihoods from CFM models. This autoencoder also has fewer parameters than the pretrained perceptual one.

Our findings suggest that autoencoder representations can be a viable solution to enhancing the reliability of likelihoods from CFM models, while maintaining the ability to generate new image data.

## 2 Related work

### 2.1 Unreliable likelihoods from normalising flows

Nalisnick et al. (2019a) were the first to observe that flow models can assign high likelihood to out-of-distribution samples. They show this for discrete-step normalising flows trained on the FashionMNIST, CIFAR10, CelebA and ImageNet datasets. Moreover, they postulate that the unreliability of likelihoods is due to the structure of the data. For instance, SVHN data has a similar mean to CIFAR10 data, and lower variance, suggesting that the distribution underlying SVHN might be contained within the distribution underlying CIFAR10. Nalisnick et al. (2019b) further relate the unreliability of likelihoods to the mismatch between a model's typical set and its areas of high probability density. Here a model's typical set refers to the set of samples whose entropy is close to the true entropy of the density (Cover & Thomas, 2012). Nalisnick et al. (2019b) also provide a statistical test to determine whether inputs are in the typical set, thereby improving the reliability of anomaly detection. Other works also consider anomaly detection from the perspective of atypicality for both discrete (Høst-Madsen et al., 2019) and continuous (Sabeti & Høst-Madsen, 2019) data, but do not consider image datasets and do not provide commentary specifically on flow models. We do not consider statistical tests for anomaly detection, but rather provide commentary on whether unmodified likelihoods from CFM models can be reliable.

Serrà et al. (2020) and Voleti et al. (2024) make observations regarding data complexity and its relation to reliable likelihoods. Serrà et al. (2020) implement a measure of input complexity through a compression algorithm that can be used for a more reliable likelihood score. Voleti et al. (2024) implement multi-resolution in continuous-time normalising flows and show that it, too, leads to unreliable likelihoods for out-of-distribution data. No solutions are provided for the continuous flows case, making our work the first to do so. Kirichenko et al. (2020) find reliable likelihoods for discrete-step normalising flows by changing the composition of the coupling layers (Dinh et al., 2015), thereby modifying the inductive biases of the flow model. By training the discrete flow model on features from a pretrained classifier, they additionally show that likelihoods become more reliable at the cost of the ability to generate data.

There are approaches that are model agnostic, or consider other types of generative models. Ren et al. (2019) observe unreliable likelihoods from autoregressive models, show that it can be attributed to background statistics, and propose a likelihood ratio method to correct for it. Choi et al. (2018) leverage the Watanabe-Akaike information criterion estimated from various generative models, including flow-based variants, to detect anomalous data. Song et al. (2019) leverage an observed difference between the training and evaluation modes of batch normalisation to identify out-of-distribution samples. Some works also demonstrate competitive out-of-distribution detection using diffusion-based models (Graham et al., 2023; Liu et al., 2023). Our work differs from theirs in that we study models that provide exact likelihoods. Zhang et al. (2023) investigate the KL divergence in flow-based models, towards an explanation of unreliable likelihoods. Ultimately, they still leverage local pixel dependencies of representations to perform anomaly detection, indicating the importance of the data representation.

Our work complements existing literature on the reliability of out-of-distribution likelihood estimates by showing that CFM models (trained with the conditional flow matching objective rather than through maximum likelihood) exhibit the same out-of-distribution behaviour. It is worth highlighting the importance of

this observation, given that CFM models are more scalable than simulation-based continuous flows. We also observe a possible relationship between unreliable likelihoods and characteristics of the data. Given previous suggestions of links between likelihood reliability and data representation (Ren et al., 2019; Serrà et al., 2020; Kirichenko et al., 2020; Zhang et al., 2023), we investigate whether autoencoder representations of image data can provide a viable solution to the likelihood reliability problem in CFM models, while preserving the ability to generate new data.

## 2.2 Representation learning

Autoencoders are a form of unsupervised representation learning, and learn feature representations from which the input can be reconstructed. Typically, the feature dimensionality is restricted in order to avoid learning an identity function between the inputs and reconstructions. These models are frequently used in self-supervised representation learning. For instance, a denoising autoencoder (Vincent et al., 2008; 2010) learns to provide clean reconstructions from noisy inputs. Vincent et al. (2008) argue that including robustness to partial destruction of the input in this way leads to learning the structure of the data manifold. Reconstructing the input under perturbation has been influential in natural language processing too, where a network is tasked to predict masked tokens (Devlin et al., 2019). Masked training objectives can be applied to images, leading to improvements in classification models that are finetuned on labelled datasets (He et al., 2022; Dong et al., 2023). Representations from autoencoders are especially useful for us, since the decoder component provides an ability for data generation from sampled feature vectors.

Other representation learning approaches could be considered when data generation is not important. In the self-supervised literature, representations can be obtained through certain pretext tasks such as context prediction (Doersch et al., 2015), solving jig-saw puzzles (Noroozi & Favaro, 2016) or rotation prediction (Gidaris et al., 2018). There are also contrastive learning approaches where a network learns a representation such that similar inputs are embedded close together, and dissimilar inputs are further apart (Chen et al., 2015; Tian et al., 2020; Khosla et al., 2020).

The use of autoencoders in generative modelling is not uncommon. Variational autoencoders (VAEs) (Kingma & Welling, 2013) and vector quantised VAEs (Van Den Oord et al., 2017), for example, are constructed in the autoencoder framework. There have also been various approaches that learn generative models in a feature space with reduced dimensionality. Vahdat et al. (2021) train score-based generative models (SGMs) on features obtained from a VAE, enabling the modelling of non-continuous data and learning of smoother SGMs in a reduced space. Rombach et al. (2022) train a diffusion model also on features from a VAE, and obtain improvements in inpainting and class-conditional image generation. We will consider their pretrained autoencoder, specifically to see if CFM models trained on the feature representations from its encoder can provide more reliable likelihoods. Dao et al. (2023) use the same autoencoder to train CFM models in a feature space with reduced dimensionality, with a specific focus on computational efficiency. We will show that this kind of pretrained autoencoder may not necessarily assist with the reliability of likelihoods. Instead, the pixel autoencoder we propose has fewer parameters and leads to more reliable likelihoods.

## 3 Methodology

We are concerned with the problem of unsupervised density estimation. Given a training set $\mathcal{D} = \{\boldsymbol{x}_i\}_{i=1}^N$ with $\boldsymbol{x} \in \mathbb{R}^d$, and defining the initial condition $\boldsymbol{z}_1 = \psi(\boldsymbol{x})$ as some feature representation of the input data, we construct a continuous flow that computes the log likelihood of $\boldsymbol{z}_1$ as

$$\log p(\boldsymbol{z}_1) = \log p(\boldsymbol{z}_0) - \int_{t_1}^{t_0} \mathrm{Tr}\left[\frac{\partial f}{\partial \boldsymbol{z}}\right] dt, \tag{2}$$

where $f_{\boldsymbol{\theta}}(\boldsymbol{z}, t)$ is the dynamics function of a (neural) differential equation defining the transformation between the data and samples from the base distribution. Following Grathwohl et al. (2019), the transformed sample $\boldsymbol{u} = \boldsymbol{z}_0$ and $\log p(\boldsymbol{z}_1)$ are obtained by simultaneously solving Equations 1 and 2 in the torchdiffeq framework (Chen, 2018) for $t \in [t_1, t_0]$. Hutchinson's trace approximation is applied to the Jacobian term

for computational efficiency. The conditional flow matching objective is a regression between $f_{\boldsymbol{\theta}}(\boldsymbol{z}, t)$ and a specified conditional vector field that generates probability paths (i.e. how the probability of a sample evolves through time). To train a CFM model, we must specify a parameterisation for the probablity path and the conditional vector field that generates it. Although more general probability paths exist (Tong et al., 2024), we restrict our focus to Gaussian conditional probability paths,

$$p_t(\boldsymbol{z} \mid \boldsymbol{z}_1) = \mathcal{N}(\boldsymbol{z} \mid \boldsymbol{\mu}(t), \sigma^2(t)\boldsymbol{I}), \tag{3}$$

where $\boldsymbol{\mu}(t)$ and $\sigma^2(t)$ describe how the mean and covariance change over time, and with $\boldsymbol{\mu}$ also dependent on $\boldsymbol{z}_1$. Such a probability path follows trajectories between a density concentrated around $\boldsymbol{z}_1$ and the base density, and is specified by a conditional vector field (Lipman et al. (2023), Theorem 3):

$$\boldsymbol{u}_t(\boldsymbol{z} \mid \boldsymbol{z}_1) = \frac{\sigma'(t)}{\sigma(t)} \left(\boldsymbol{z} - \boldsymbol{\mu}(t)\right) + \boldsymbol{\mu}'(t), \tag{4}$$

where the prime symbol indicates the derivative with respect to $t$. By defining $\boldsymbol{\mu}(t) = t\boldsymbol{z}_1$ and $\sigma(t) = 1 - (1 - \sigma_{\min})t$, the target conditional vector field leading to the standard Gaussian base becomes

$$\boldsymbol{u}_t(\boldsymbol{z} \mid \boldsymbol{z}_1) = \frac{\boldsymbol{z}_1 - (1 - \sigma_{\min})\boldsymbol{z}}{1 - (1 - \sigma_{\min})t}. \tag{5}$$

The conditional vector field $\boldsymbol{u}_t(\boldsymbol{z} \mid \boldsymbol{z}_1)$, with $\sigma_{\min}$ set sufficiently small, leads to the conditional flow matching objective (Lipman et al., 2023):

$$\mathcal{L}(\boldsymbol{\theta}) = \frac{1}{N} \sum_{i=1}^{N} ||f_{\boldsymbol{\theta}}(\boldsymbol{z}, t) - \boldsymbol{u}_t(\boldsymbol{z} \mid \boldsymbol{z}_1)||^2, \tag{6}$$

which is an average over $N$ samples in a mini-batch, and with probability paths defined over $t \sim \mathcal{U}(0, 1)$. After training, we may compute likelihoods by first obtaining the feature representation $\boldsymbol{z}_1 = \psi(\boldsymbol{x})$ and then solving Equation 2. New samples of images can be generated by the model, if the inverse $\psi^{-1}$ can be evaluated. To do so, we first sample $\boldsymbol{z}_0$ from the base distribution of the trained CFM model, and solve the differential equation in the reverse direction to obtain $\boldsymbol{z}_1$ (i.e. from time $t_0$ to $t_1$). The corresponding image sample can then be obtained by computing $\boldsymbol{x} = \psi^{-1}(\boldsymbol{z}_1)$.

Our goal is to determine whether there are parameterisations of $\psi(\boldsymbol{x})$ that may lead to reliable likelihoods from the corresponding CFM model. We consider three variants, as described in the sections below. For experimental evaluation we consider the MNIST, FashionMNIST, CIFAR10 and SVHN datasets.

## 3.1 Features from EfficientNet-B4

As a starting point, we consider the EfficientNet-B4 network pretrained on ImageNet. 1792-dimensional features are extracted from one of the fully connected layers of this model. We thereby reproduce the experimental procedure of Kirichenko et al. (2020), but for continuous flows.

To investigate the impact of presenting this pretrained network with data outside of its training data distribution, we trained a four-class LDA classifier on the EfficientNet-B4 features obtained from the MNIST, FashionMNIST, CIFAR10, and SVHN datasets, where we treat these datasets as the four classes, and achieved a test classification accuracy of 99%. This provides some evidence that the features from EfficientNet-B4 do differentiate at the dataset level, and suggests that a model learning the density over one of these datasets, in the EfficientNet-B4 feature space, may discern feature vectors from the other datasets as out-of-distribution. We also show in Figure 2 that there is a sense of separation between the datasets, even for a 2-dimensional LDA projection of the 1792-dimensional feature vectors.

We note that the conversion from an image to this feature representation is not invertible, preventing the generation of image samples from a CFM model trained in this feature space.

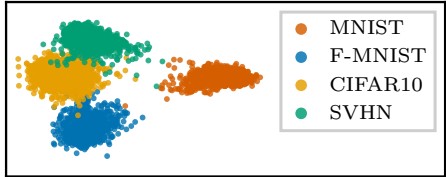

Figure 2: Two-dimensional LDA projections of the EfficientNet-B4 feature representations of training samples from MNIST, FashionMNIST (F-MNIST), CIFAR10 and SVHN.

## 3.2 Features from a perceptual autoencoder

To retain the ability to generate image samples, we also consider feature vectors obtained from a pretrained autoencoder (Rombach et al., 2022). This autoencoder was trained using a perceptual distance loss (Zhang et al., 2018) that measures the perceptual similarity between two images, and an adversarial objective (Larsen et al., 2016; Isola et al., 2017; Dosovitskiy & Brox, 2016; Esser et al., 2021; Yu et al., 2022) based on image patches (Isola et al., 2017). This combination of objectives is motivated by evidence that an objective based only on pixel reconstruction is inadequate in terms of decoding high-quality images (Wang & Bovik, 2009; Larsen et al., 2016). For instance, Wang & Bovik (2009) show with an example that various image distortions can lead to the same mean squared error when compared to the original image.

The original purpose of this autoencoder was perceptual image compression for computational efficiency in diffusion-based generative models. We instead focus on whether an autoencoder primed for high-quality image reconstruction can assist with reliable likelihoods from CFM models, and provide a means of generating new images of high quality.

We note that feature representations from this perceptual autoencoder have a shape of (28, 28, 4), and is larger than the original image dimensions of our datasets which are either (28, 28, 1) or (32, 32, 3). The pixel autoencoder we describe in Section 3.3 will reduce the input dimensionality.

Figure 3 (top) shows the feature representations and reconstructions of randomly chosen CIFAR10 samples, obtained from the perceptual autoencoder. Examples from the other datasets can be found in Appendix A. It is apparent that the reconstructions for these datasets, on which the autoencoder was not trained, are of high quality. It seems that the autoencoder has learned a general purpose feature representation. That the feature representations are perceptually similar to the input is striking, and reinforces that the main purpose of this autoencoder is perceptual image compression. A similar observation followed after we trained the autoencoder with a feature dimensionality of (16, 16, 1), as illustrated in the bottom row of Figure 3.

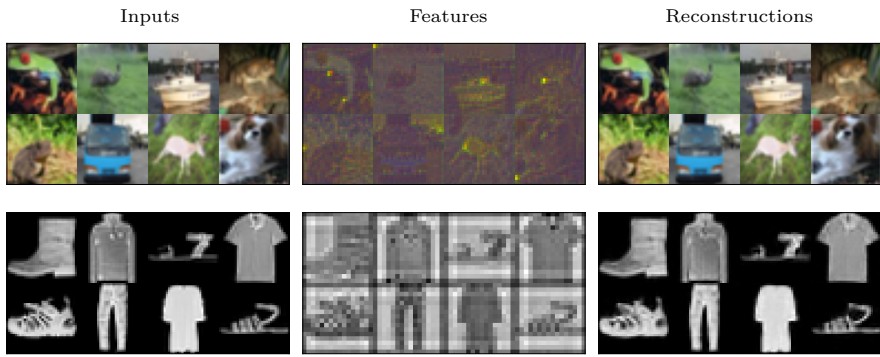

Figure 3: Top row: images from CIFAR10, gamma-corrected feature representations from the pretrained perceptual autoencoder, and decoded reconstructions. Bottom row: images from FashionMNIST, features from a trained perceptual autoencoder in which the feature dimensionality is smaller than the input, and decoded reconstructions. We observe that these feature spaces are perceptually similar to the input images.

We found that a fully convolutional autoencoder trained with a pixel-based reconstruction loss also retains perceptual similarity in the feature space, suggesting that the perceptual loss might not be the cause of this observation.

### 3.3 Features from a pixel autoencoder

As an alternative to the perceptual autoencoder, which appears to retain some perceptual similarity in its features, we consider a simpler autoencoder with a pixel-based, mean squared error reconstruction objective. This autoencoder reduces the dimensionality of the data, which will assist with computational efficiency. It has the advantage of a significantly reduced model capacity compared to the perceptual autoencoder, which can act as a form of regularisation and lead to faster training. Despite its reduced capacity, the autoencoder can offer sharp reconstructions under suitable hyperparameters.

The encoder module consists of 5 convolutional layers, 3 of which are strided, followed by a flattening operation and a fully connected layer that controls the dimensionality of the feature space and assists with suppressing perceptual similarity in the encoded features. The decoder module consists of a fully connected layer, followed by a reshaping operation and 5 convolutional layers that attempt to reconstruct the input to the encoder. The autoencoder is trained for 500 epochs, with the Adam optimiser (Kingma & Ba, 2015) and a learning rate scheduler that reduces the learning rate from 0.001 when the validation loss plateaus.

Figure 4 shows example feature representations and reconstructions for images from the CIFAR10 and SVHN datasets, obtained from a pixel autoencoder trained on CIFAR10. For display purposes, we reshaped the 768-dimensional feature vectors obtained after the encoder's fully connected layer to images of size (16, 16, 3). We note that perceptual similarity does not seem to be preserved in the features. Reshaping the features in this way also allows for a U-Net parameterisation of the vector field (Ho et al., 2020; Lipman et al., 2023), which seems to significantly impact the convergence of CFM model training. We further note that the quality of the reconstructions is high, even for images from datasets other than the autoencoder's training set, indicating that the model has learned a general purpose feature representation. Examples of feature representations and reconstructions for images from the MNIST and FashionMNIST datasets can be found in Appendix B.

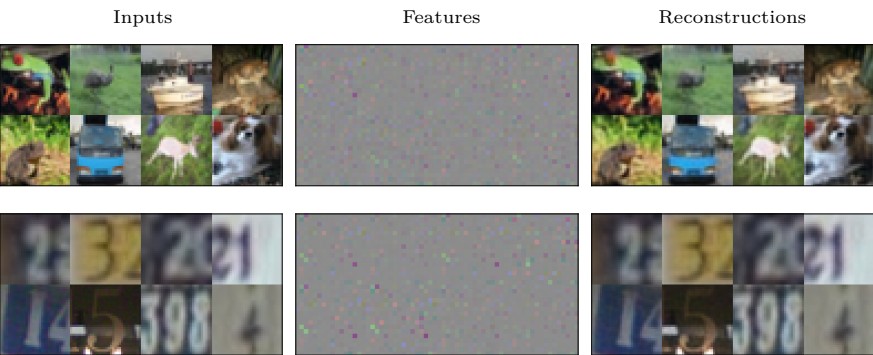

Figure 4: Input images (left), reshaped feature representations (middle), and reconstructions (right) from the pixel autoencoder trained on the CIFAR10 dataset. Perceptual similarity to the input is no longer evident in the features, and the decoder produces good reconstructions even when presented with inputs from the SVHN dataset (bottom row).

### 3.4 CFM model parameterisation

We train CFM models on each of the respective feature representations presented above, and will compare their performance against baseline CFM models trained in the original image space. Table 1 provides a summary.

The vector field for CFM models trained on EfficientNet-B4 (ENet) features is parameterised by a fully connected network with skip connections. We experimented with a sinusoidal time embedding and also a hypernetwork for time dependence, but found that adding time as an additional input dimension worked better. The vector field for the baseline, perceptual autoencoder (PercAE) features, and pixel autoencoder (PixAE) features is parameterised using the same time-dependent U-Net with attention, modified according to the input feature dimensionality. Further implementation details are included in Appendix C.

Table 1: Summary of the respective feature representations on which we train CFM models. Greyscale and colour inputs are denoted by (g) and (rgb). PixAE is different from PercAE in that it contains fully connected layers to distort the representations, and is trained only with a pixel-based reconstruction loss. The last column indicates whether or not the trained CFM model can be used to generate images.

| Feature space | Description | Pretrained | Feature vector length | Image gen. |
|---|---|---|---|---|
| Baseline | Original image space | ✗ | 784 (g), 3072 (rgb) | ✓ |
| ENet | Features from an EfficientNet-B4 classifier | ✓ | 1792 | ✗ |
| PercAE | Autoencoder trained with a perceptual loss | ✓ | 3136 | ✓ |
| PixAE | Autoencoder trained with a pixel-based loss | ✗ | 256 (g), 768 (rgb) | ✓ |

## 3.5 Evaluation metrics

**Likelihood metric.** Bits-per-dimension, derived from the average log likelihood, is a common metric for measuring generalisation in likelihood-based generative models of discrete image data (Papamakarios et al., 2017). Bits-per-dimension is not applicable to our feature representations of the data, as it would require infinite bits to encode continuous data under the model. Instead we use a signed version of the Bhattacharyya distance for out-of-distribution analysis. Given two Gaussian densities $h_1 = \mathcal{N}\left(\mu_1, \sigma_1^2\right)$ and $h_2 = \mathcal{N}\left(\mu_2, \sigma_2^2\right)$, the signed distance is defined as

$$D_{\text{SB}}\left(h_1, h_2\right) = \text{sign}(\mu_2 - \mu_1)\left[\frac{1}{4}\frac{\left(\mu_2 - \mu_1\right)^2}{\sigma_2^2 + \sigma_1^2} + \frac{1}{2}\log\left(\frac{\sigma_2^2 + \sigma_1^2}{2\sigma_2\sigma_1}\right)\right], \tag{7}$$

where we set $\text{sign}(0) = 1$ to avoid a distance of zero for equal means. $D_{\text{SB}}\left(h_1, h_2\right)$ evaluates to a high value when the means of $h_1$ and $h_2$ are far apart, and their standard deviations are small, thereby indicating less overlap between the densities. $D_{\text{SB}}\left(h_1, h_2\right)$ is positive when $\mu_2 \geq \mu_1$, indicating that $h_2$ is shifted rightwards from $h_1$, and it is negative when $\mu_2 < \mu_1$. The signed Bhattacharyya distance between two Gaussians with equal parameters is 0. In our evaluations, a distance will be obtained by fitting Gaussian densities to in- and out-of-distribution likelihood histograms, and measuring the magnitude and direction of their overlap. A large magnitude indicates that there is little overlap between the two densities. A positive sign indicates that in-distribution log likelihoods are, on average, higher than out-of-distribution log likelihoods. Large positive distances are therefore indicative of reliable likelihoods.

**Sample quality.** To evaluate sample quality from a trained CFM model, we will make use of the Fréchet inception distance (FID) (Heusel et al., 2017) between the training set and 50K generated samples. For the CFM models trained on ENet features, the LDA classifier from before is used to inspect samples of generated feature vectors. These generated feature vectors cannot be converted to images, and the classifier scores do not describe sample quality. We consider the LDA classifier merely to verify sensible output from the model.

Quantitative metrics are calculated over 5 training runs, and specific hyperparameters and training configurations are provided in Appendix C. We separate our results into three sections, for the three feature spaces, and compare each with a baseline CFM model trained on the original images. A holistic comparison between all three approaches is made at the end. Code for reproducing the results will be made available upon acceptance of this paper.

# 4 Results

## 4.1 CFM models trained on ENet features

Figures 5a and 5b show log likelihood histograms from baseline and ENet models trained on FashionMNIST. The CFM model trained on ENet features assigns slightly lower likelihoods on average to out-of-distribution data, compared to the baseline. There is still considerable overlap in the histograms of in- and out-of-distribution likelihoods, and room for improvement. Figures 5c and 5d show similar (though somewhat worse) behaviour for a CFM model trained on ENet features of CIFAR10. The impact on likelihood reliability that we observe here, by training CFM models on ENet feature representations, is not as significant for the CIFAR10 dataset as in the reported results for discrete-step flow models. This may be due to the continuous formulation of our models, or due to the parameterisation of the vector field. Where we use a fully connected network with skip connections, Kirichenko et al. (2020) incorporate their own st-network for the discrete flow. We tried other methods of incorporating time dependence, such as a hypernetwork (Ha et al., 2017), but did not find any improvements.

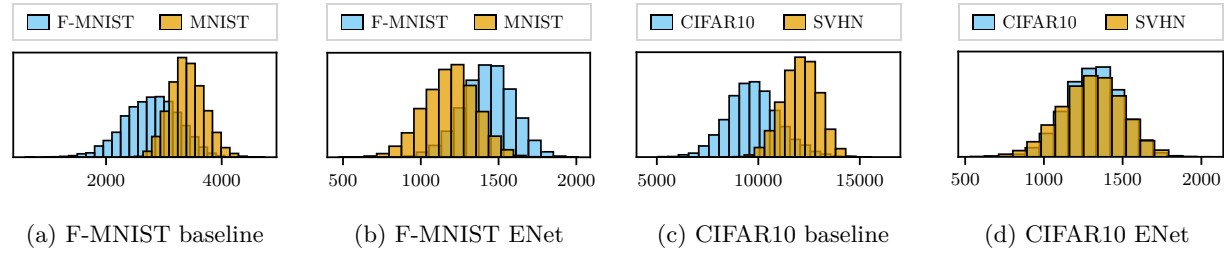

(a) F-MNIST baseline      (b) F-MNIST ENet      (c) CIFAR10 baseline      (d) CIFAR10 ENet

Figure 5: Log likelihood histograms from baseline CFM models trained on original images, and from CFM models trained on **ENet features** of FashionMNIST and CIFAR10. Blue corresponds to in-distribution test data, and orange to out-of-distribution data.

Table 2 lists the signed Bhattacharyya distances between various in- and out-of-distribution log likelihood histograms. A positive value indicates that the mean of in-distribution likelihoods is higher than the mean of out-of-distribution likelihoods. Models trained on the MNIST and SVHN datasets provide reliable likelihoods, as implied by the positive distances in Table 2. The results in the table also provide quantitative evidence for the improvements seen in Figure 5. We conclude that a CFM model can benefit from training on EfficientNet-B4 feature representations, corroborating what has been shown for discrete-step normalising flows. However, since the conversion of images to this kind of feature representation is not invertible, the resulting CFM models cannot be used to generate new samples. As a basic test of sample quality, we show in Appendix D that generated feature vectors are discernable by the LDA classifier from Section 3.1.

Table 2: Signed Bhattacharyya distances between log likelihood histograms of in- and out-of-distribution data, from CFM models trained on original images (the baselines) and on **ENet features**, with in- and out-of-distribution sets as shown. Means and standard deviations are measured over 5 training runs.

| In-distribution | Out-of-distribution | Baseline | ENet features |
|---|---|---|---|
| MNIST | F-MNIST | $2.13 \pm 0.06$ | $3.55 \pm 0.43$ |
| F-MNIST | MNIST | $-0.85 \pm 0.01$ | $0.74 \pm 0.02$ |
| CIFAR10 | SVHN | $-1.08 \pm 0.00$ | $-0.30 \pm 0.40$ |
| SVHN | CIFAR10 | $1.72 \pm 0.02$ | $2.49 \pm 0.51$ |

## 4.2 CFM models trained on PercAE features

Figures 6a and 6b show log likelihood histograms for the baseline and PercAE models trained on Fashion-MNIST. We observe that a CFM model trained on PercAE features can provide more reliable likelihood estimates for out-of-distribution data from MNIST. However, when trained on CIFAR10, both models assign

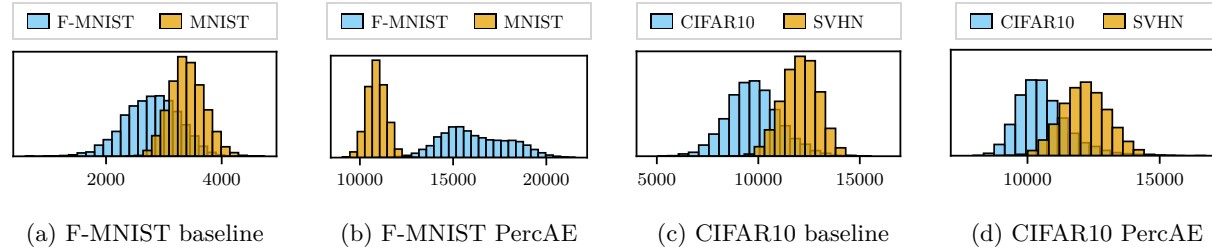

(a) F-MNIST baseline     (b) F-MNIST PercAE     (c) CIFAR10 baseline     (d) CIFAR10 PercAE

Figure 6: Log likelihood histograms from baseline CFM models trained on original images, and from CFM models trained on **PercAE features** of FashionMNIST and CIFAR10.

higher likelihoods to out-of-distribution data from SVHN, as shown in Figures 6c and 6d. Models trained on MNIST and SVHN exhibit similar trends to those reported in Table 2, correctly assigning lower likelihoods to FashionMNIST when trained on MNIST and to CIFAR10 when trained on SVHN. Table 3 shows the signed Bhattacharyya distances between in- and out-of-distribution log likelihood histograms from CFM models trained on PercAE features.

For the CIFAR10 dataset, training on PerAE features appears to lead to less reliable likelihoods compared to ENet features. The increased dimensionality of PercAE features may not be the cause, as both ENet and PercAE features are high in dimension yet lead to reliable likelihoods from models trained on FashionMNIST. Instead, it may be that the ENet feature space better separates the datasets we are considering, due to it being pretrained to recognise different object classes. Additionally, the perceptual similarities between input images and their PercAE feature representations (as demonstrated in Figure 3) is somewhat conspicuous, and may point towards an explanation for not seeing an improvement in likelihood reliability for a dataset like CIFAR10 that require more a complex feature representation in order for a CFM model to effectively capture the underlying distribution.

Figure 7 shows generated features and decoded images from CFM models trained on the PercAE features of FashionMNIST and CIFAR10 which, qualitatively, appears to be quite good.

FashionMNIST features     FashionMNIST reconstructions     CIFAR10 features     CIFAR10 reconstructions

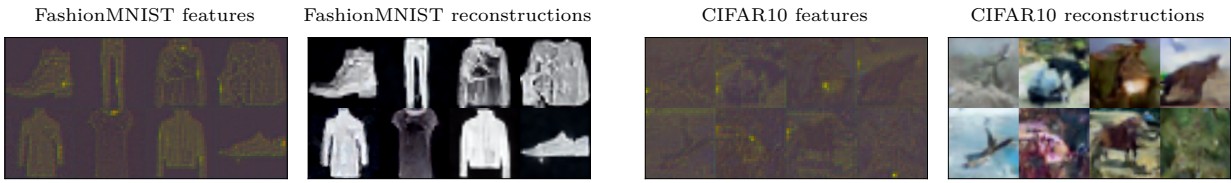

Figure 7: Examples of feature vectors generated by CFM models trained on **PercAE features** of Fashion-MNIST (left) and CIFAR10 (right), and the decoded reconstructions. We show gamma-corrected versions of the first three channels of the feature vectors.

### 4.3 CFM models trained on PixAE features

Figure 8 shows log likelihood histograms from baseline and PixAE models trained on FashionMNIST and CIFAR10. The baseline CFM models again assign higher likelihoods to out-of-distribution data. However, the models trained on PixAE features provide clearly separated likelihoods for in- and out-of-distribution data, with lower likelihoods for the latter, across all datasets. Table 3 shows the signed Bhattacharyya distances between in- and out-of-distribution likelihoods, from CFM models trained on PixAE features, compared to the baseline models and models trained on ENet and PercAE features. The consistently positive distances for the CFM models trained on PixAE features, across all datasets, again indicate an improvement in likelihood reliability compared to the other feature spaces. The pixel autoencoder also has fewer parameters than the alternatives we considered.

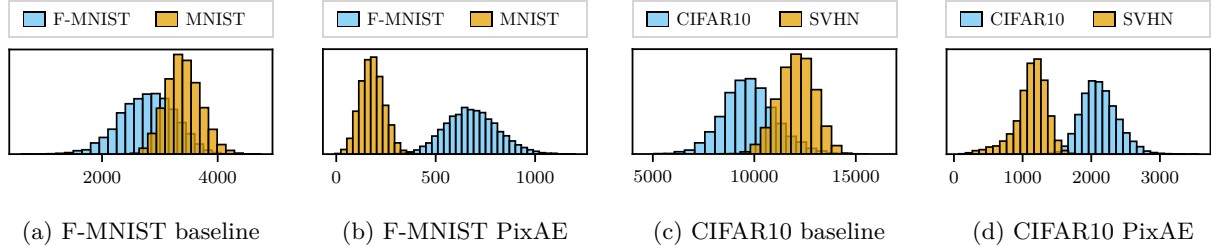

(a) F-MNIST baseline  (b) F-MNIST PixAE  (c) CIFAR10 baseline  (d) CIFAR10 PixAE

Figure 8: Log likelihood histograms from baseline CFM models trained on original images, and from CFM models trained on **PixAE features** of FashionMNIST and CIFAR10.

Table 3: Signed Bhattacharyya distances between log likelihood histograms of in- and out-of-distribution data, from CFM models trained on original images (the baselines), ENet features, PercAE features, and PixAE features. Means and standard deviations are measured over 5 training runs.

| In-distribution | Out-of-distribution | Baseline | ENet features | PercAE features | PixAE features |
|---|---|---|---|---|---|
| MNIST | F-MNIST | $2.13 \pm 0.06$ | $3.55 \pm 0.43$ | $14.25 \pm 2.28$ | $5.60 \pm 0.13$ |
| F-MNIST | MNIST | $-0.85 \pm 0.01$ | $0.74 \pm 0.02$ | $4.06 \pm 0.77$ | $3.90 \pm 0.19$ |
| CIFAR10 | SVHN | $-1.08 \pm 0.00$ | $-0.30 \pm 0.40$ | $-0.89 \pm 0.00$ | $2.67 \pm 0.08$ |
| SVHN | CIFAR10 | $1.72 \pm 0.02$ | $2.49 \pm 0.51$ | $2.00 \pm 0.13$ | $4.41 \pm 0.10$ |

We show qualitatively in Figure 9 that features generated by our trained CFM models can successfully be decoded to images, but note that samples from the PixAE models trained on CIFAR10 and SVHN are rather low in quality. There might be a trade-off: PercAE features lead to good samples but some inconsistency in the reliability of likelihood estimates, while PixAE features lead to more reliable likelihoods but poorer samples.

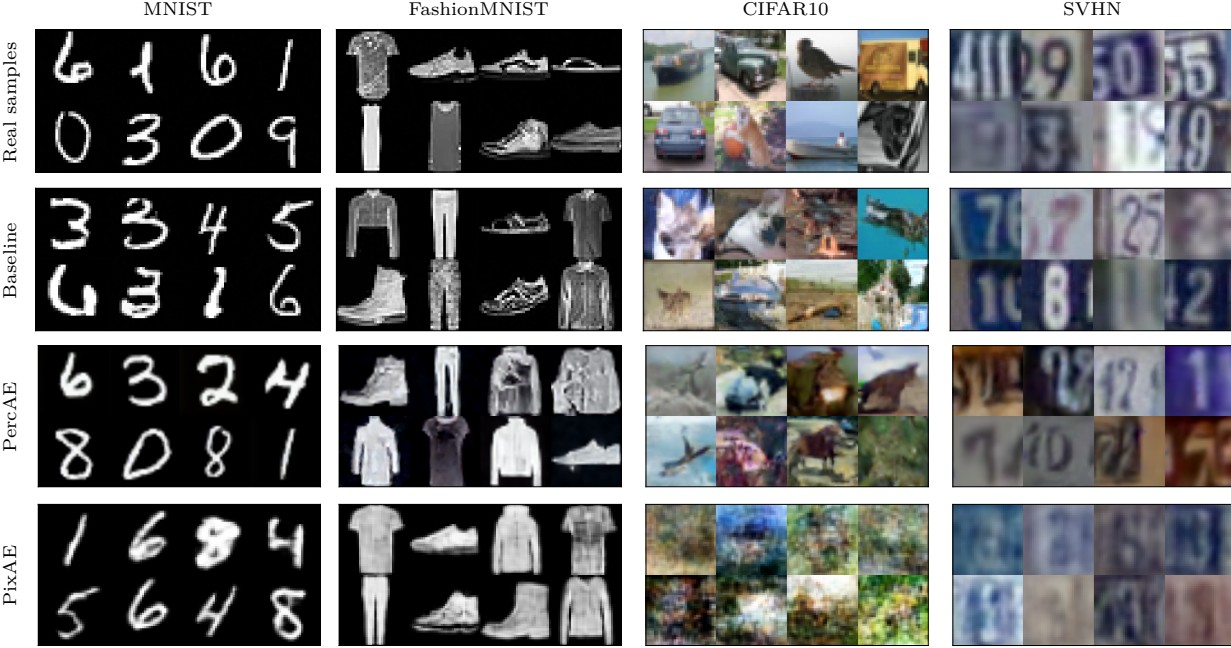

Figure 9: Real samples and decoded features generated from CFM models trained on baseline features (the original pixels), PercAE features and PixAE features.

Table 4 confirms our qualitative assessment, where we see that FID scores for generated samples from the PixAE models are relatively high compared to the others. For context, Lipman et al. (2023) report FID scores of around 8.0 for other generative models trained on CIFAR10. We experimented with training the CFM models for longer, but it did not improve sample quality. The problem persisted for greyscale versions of CIFAR10 and SVHN, indicating that the complexity through colour channels is not the main cause. The high standard deviation in the baseline model trained on SVHN is due to one training run that produced particularly bad samples (with low quality in terms of realism, or low diversity in the samples, or both).

We also experimented with training CFM models on CIFAR10 features obtained from a "hybrid" autoencoder, with the architecture of our pixel autoencoder but trained from scratch with the perceptual loss, and measured a slightly better FID score of $114.82 \pm 1.08$. A few samples are shown in Figure 10. This model also provides reliable likelihoods on out-of-distribution data from SVHN, with a Bhattacharyya distance of $1.03 \pm 0.03$. We further experimented with training CFM models on features obtained from autoencoders with fully convolutional architectures, and again obtained samples of high quality but unreliable likelihoods. The interplay between sample quality, likelihood reliability, and structure in the autoencoder feature space offers promising avenues for future research.

Table 4: Fréchet inception distances of generated samples from CFM models trained on baseline features (the original pixels), PercAE features, and PixAE features of the respective datasets. Means and standard deviations are measured over 5 training runs.

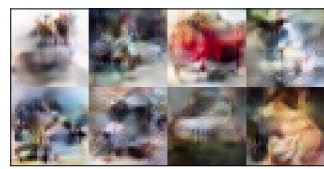

Figure 10: Samples generated by a CFM model trained on CIFAR10 features from a hybrid autoencoder.

|  | MNIST | F-MNIST | CIFAR10 | SVHN |
|---|---|---|---|---|
| Baseline | $3.20 \pm 1.25$ | $5.37 \pm 0.83$ | $27.62 \pm 1.78$ | $33.10 \pm 39.85$ |
| PercAE | $29.89 \pm 2.99$ | $49.32 \pm 3.64$ | $56.67 \pm 0.25$ | $38.87 \pm 2.62$ |
| PixAE | $31.48 \pm 1.41$ | $58.02 \pm 4.08$ | $190.30 \pm 5.90$ | $104.82 \pm 9.12$ |

# 5 Conclusion

We investigated the tendency of continuous-time normalising flows trained with the conditional flow matching objective to assign undesirably high likelihoods to out-of-distribution data, and whether training in feature space can improve matters. We evaluated three feature representations, and found that those from a pretrained classifier and a pretrained perceptual autoencoder could improve likelihood reliability only for some datasets, while the features from our pixel autoencoder provided consistent improvements. Autoencoder features address the limitation of classifier-derived features by maintaining the ability to generate image samples. However, samples from models trained on features from our pixel autoencoder were found to be somewhat lower in quality compared to those from the pretrained perceptual autoencoder. The latter, on the other hand, did not fully resolve the likelihood reliability problem. We hypothesise that introducing structural biases, such as class label information, could enhance sample quality for datasets like CIFAR10 and SVHN. Further validation on datasets like CIFAR100, CelebA, and ImageNet is also needed, once image generation capability improves. The impact of perceptual similarity between input images and their feature representations on likelihood reliability remains open for further investigation.

Our findings contribute to the discourse on improving likelihood reliability in scalable continuous-time flow models, and highlight the role of feature representations in addressing this challenge.

**Broader impact statement**

Unreliable likelihoods from flow-based generative models may foster a false sense of confidence in their predictions. This mistaken equivalence between high confidence and model accuracy can result in poor decision making in critical or high-risk fields such as healthcare (e.g. diagnosing medical conditions), finance (e.g. approving loans), and security (e.g. detecting fraud). Asserting the reliability of generative models can accelerate their adoption but also heighten the risk of misuse. Therefore, we advocate for continued research into the reliability of flow-based generative models.

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

## A  Features and reconstructions from the perceptual autoencoder

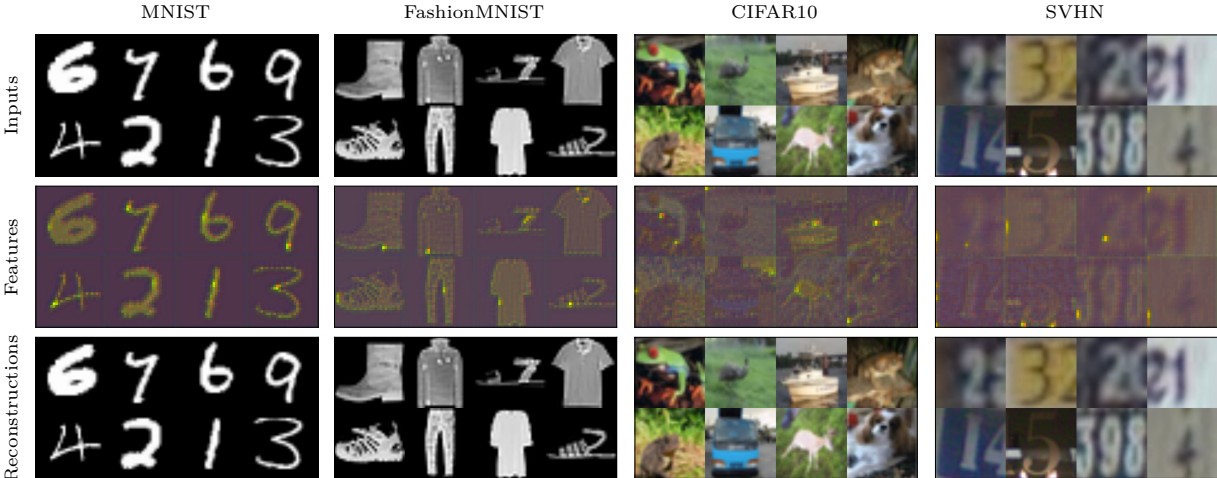

Figure 11: Example images, gamma-corrected feature representations and reconstructions from the pre-trained perceptual autoencoder described in Section 3.2, for the MNIST, FashionMNIST, CIFAR10 and SVHN datasets. Perceptual similarity between the features and the input images is evident.

## B  Features and reconstructions from the pixel autoencoder

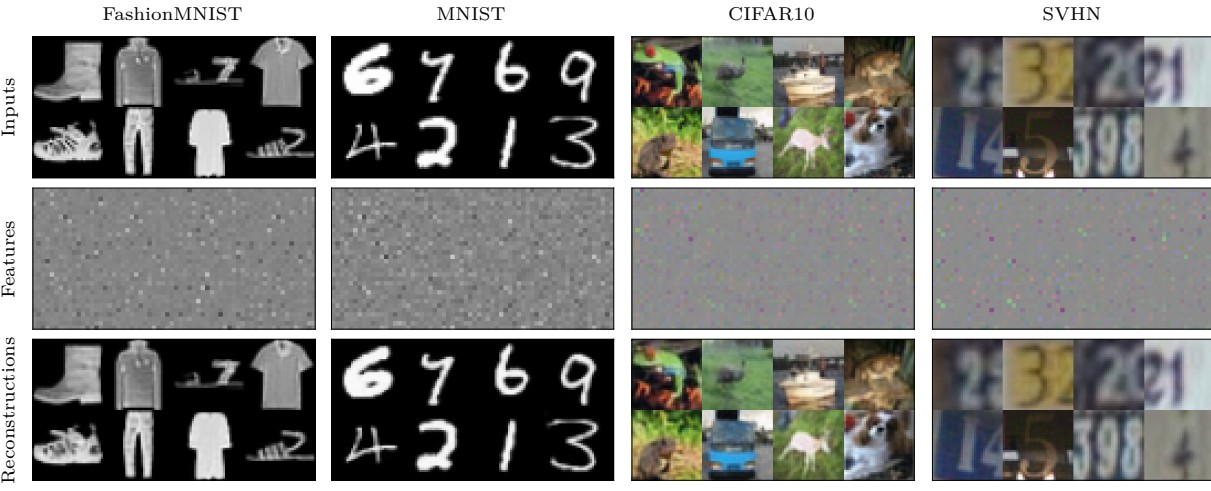

Figure 12: Example images, reshaped feature vectors and reconstructions from two versions of the pixel autoencoder described in Section 3.3. Perceptual similarity to the input is no longer evident in the feature space, and the decoder provides good reconstructions even when the autoencoder is presented with input images from a dataset different to its training set.

# C Implementation details

Log likelihoods and generated samples from all models are computed using the torchdiffeq framework (Chen, 2018) in PyTorch. All models are trained on a single NVIDIA RTX A6000 GPU. The Adam optimiser (Kingma & Ba, 2015) is used with default values for $\beta_1$ and $\beta_2$, and its learning rate is warmed up with a linear scheduler. Hyperparameter tuning was performed on the learning rate and number of epochs, to the limits of our compute budget.

The implementation from Tong et al. (2024) is adapted for CFM models trained on the original images, on features from the perceptual autoencoder, and on features from the pixel autoencoder. We refer the reader to the original implementation of Tong et al. (2024) for descriptions of the various hyperparameters. Final hyperparameter values for each CFM model are provided in the following sections.

## C.1 CFM models trained on the original images

The vector field for the baseline CFM models trained on original images uses the hyperparameters shown in Table 5.

Table 5: Hyperparameters for the baseline CFM models trained on original image data.

| Parameter | MNIST | FashionMNIST | CIFAR10 | SVHN |
|---|---|---|---|---|
| Channels | 128 | 128 | 128 | 128 |
| Channels multiple | (1, 2, 2) | (1, 2, 2) | (1, 2, 2, 2) | (1, 2, 2, 2) |
| Heads | 1 | 1 | 1 | 1 |
| Heads channels | 1 | 1 | 1 | 1 |
| Attention resolution | 16 | 16 | 16 | 16 |
| Dropout | 0.0 | 0.0 | 0.0 | 0.0 |
| Batch size | 128 | 256 | 256 | 256 |
| Epochs | 150 | 150 | 150 | 150 |
| Learning rate (warmed up) | 0.0002 | 0.0002 | 0.0002 | 0.0002 |

## C.2 CFM models trained on EfficientNet-B4 features

The vector field for CFM models trained on EfficientNet-B4 features is parameterised by a fully connected neural network with skip connections. We append time as an input, use 5 hidden layers, and skip connections between hidden layers 1 and 5 and hidden layers 2 and 4. The Swish activation function is used (Ramachandran et al., 2017). Table 6 shows additional hyperparameters for these CFM models.

Table 6: Hyperparameters for CFM models trained on features obtained from EfficientNet-B4.

| Parameter | MNIST | FashionMNIST | CIFAR10 | SVHN |
|---|---|---|---|---|
| Number of hidden layers | 5 | 5 | 5 | 5 |
| Layer width | 1610 | 1610 | 1610 | 1610 |
| Batch size | 1024 | 1024 | 1024 | 1024 |
| Epochs | 300 | 300 | 300 | 300 |
| Learning rate (warmed up) | 0.00005 | 0.00005 | 0.00005 | 0.00005 |

## C.3 CFM models trained on perceptual autoencoder features

The vector field for CFM models trained on perceptual autoencoder features uses a modified version of the U-Net parameterisation listed for the baseline, since inputs are of different dimensionality. Table 7 lists the hyperparameters.

Table 7: Hyperparameters for CFM models trained on features from the perceptual autoencoder.

| Parameter | MNIST | FashionMNIST | CIFAR10 | SVHN |
|---|---|---|---|---|
| Channels | 128 | 128 | 128 | 128 |
| Channels multiple | (1, 2, 2) | (1, 2, 2) | (1, 2, 2, 2) | (1, 2, 2, 2) |
| Heads | 1 | 1 | 1 | 1 |
| Heads channels | 1 | 1 | 1 | 1 |
| Attention resolution | 16 | 16 | 16 | 16 |
| Dropout | 0.0 | 0.0 | 0.0 | 0.0 |
| Batch size | 128 | 128 | 128 | 128 |
| Epochs | 100 | 100 | 100 | 100 |
| Learning rate (warmed up) | 0.0002 | 0.0002 | 0.0002 | 0.0002 |

### C.4 CFM models trained on pixel autoencoder features

The architecture of the pixel autoencoder is described in Section 3.3. A stride length of 2 is used in each of the strided convolutions in the encoder, and in each of the transposed convolutions in the decoder. A kernel size of 3 is used throughout, and the Gaussian error linear unit (GELU) activation function (Hendrycks & Gimpel, 2016) is used. Table 8 lists the modified hyperparameters for CFM models trained on these pixel autoencoder features.

Table 8: Hyperparameters for CFM models trained on features from the pixel autoencoder.

| Parameter | MNIST | FashionMNIST | CIFAR10 | SVHN |
|---|---|---|---|---|
| Channels | 128 | 128 | 128 | 128 |
| Channels multiple | (1, 2, 2, 2) | (1, 2, 2, 2) | (1, 2, 2, 2) | (1, 2, 2, 2) |
| Heads | 1 | 1 | 1 | 1 |
| Heads channels | 1 | 1 | 1 | 1 |
| Attention resolution | 16 | 16 | 16 | 16 |
| Dropout | 0.0 | 0.0 | 0.0 | 0.0 |
| Batch size | 128 | 128 | 128 | 128 |
| Epochs | 100 | 100 | 100 | 100 |
| Learning rate (warmed up) | 0.0002 | 0.0002 | 0.0002 | 0.0002 |

## D  LDA classifier on EfficientNet-B4 features

Table 9 reports the LDA classifier accuracy on feature vector samples generated by CFM models trained on EfficientNet-B4 features of MNIST, FashionMNIST, CIFAR10, and SVHN, respectively. This provides some verification that the four trained CFM models can generate feature vectors that are relatively close to their respective training sets. We suspect that the accuracy can be increased through further hyperparameter tuning.

Table 9: LDA classifier accuracy on samples generated by seperate CFM models trained on EfficientNet-B4 features of MNIST, FashionMNIST, CIFAR10 and SVHN.

| | MNIST | FashionMNIST | CIFAR10 | SVHN |
|---|---|---|---|---|
| LDA accuracy | 0.8419 | 0.8452 | 0.8586 | 0.7660 |

