# OpenReview forum: "Reliable likelihoods from conditional flow matching generative models in feature space"
_TMLR — Rejected by TMLR_

### Review · Reviewer_zWos · 2024-10-18

**Summary Of Contributions:**

The paper studies OOD likelihoods obtained from Conditional Flow (or Diffusion) models. The paper trains the CFM model in four different ways: directly on pixels and features from perceptual autoencoder (PercAE), efficientnet and pixel autoencoder (PixAE). Their major insights are as follows:

* CFM models trained on pixels cannot determine OOD inputs reliably (Fig 5a and Fig 5c) similar to discrete normalizing flows.
* The paper compares CFM models trained on different feature representations.and obtain the following ranking in terms of effectiveness: 1) PercAE features (not very effective) 2) Efficient Nets (slightly effective) and 3) PixelAE features (very effective).
* The features of PercAE exhibit some low-level image-specific structure which is absent in PixelAE. So the authors hypothesize that this property leads to worse OOD detection of PercAE.

**Audience:**

Yes

**Broader Impact Concerns:**

Nope

**Claims And Evidence:**

No

**Requested Changes:**

* There have been at least a couple of works DDPM-OOD (https://arxiv.org/pdf/2211.07740) and LMD (https://arxiv.org/abs/2302.10326) that show diffusion models can reliably be used for out-of-detection. DDPM-OOD shows that this can be done with diffusion models trained in pixel space while this paper offers contradictory results. The paper can contextualize their work with these other techniques with some discussion. It is even better if they compare their approach with PixelAE features to DDPM-OOD on some of the datasets. For example, the authors can report the AUC score they obtain and compare it to other methods from Table 1 of DDPM-OOD.

* The authors train a baseline CFM model directly on pixels. However, the quality of this model is unclear to me so the authors may report the FID of this baseline. It does not have to be SOTA but it should be convincing that this model can generate high-quality images.

* Table 2 is also a bit unsatisfactory. For example, the LDA classifier can reliably predict only 75% of time that a predicted sample is from SVHN. This raises concerns on how good the CFM model on EfficientNet features is.

* The authors can similarly report FID on samples generated using the CFM on PercAE features and reconstructed with the decoder.

* In the introduction, the paper can add a bit more high-level motivation on studying CFM’s for out-of-distribution detection. For example, it is now known that normalizing flows do not provide reliable likelihoods. Why would  the results/insights to be different as compared to discrete normalizing flows?

### Compressing image-specific structures leads to better OOD detectability.

The authors hypothesize that absence of low-level image structure may lead to better OOD detection.

* It seems to me that this is more of correlation than causation. The paper may soften the claim with regards to this.
* The final layer of EfficientNet may also lack low-level image structure but it performs worse on OOD as compared to PixelAE. So there are other factors that contribute to OOD detection. The authors can add some intuition on this in the paper.
* I assume that the features used are from the pooling layer of EffNet. The paper can train CFM Model on earlier feature representations above the pool layer which would retain more image-specific structure as compared to the pooling later. If the OOD detectability of these layers are worse than the pooling layer, this will add further evidence to this hypothesis.
* More information on the PixelAE architecture is needed. For example
    * How does the architecture go from 2-D spatial representation to 1-D representation?
    * If the bottleneck representation has no spatial information then how does the model reconstruct visually perfect images from this bottleneck? Are there skip connections from the downsampling blocks to the upsampling blocks?

**Strengths And Weaknesses:**

### Strengths:

* The paper is easy to read
* The results show that CFM models trained on PixelAE features leads to better OOD detection. This is quite interesting.


### Weaknesses
My major comments are on contextualing the work given prior literature, the quality of the CFM models used in the study and the claim that compressing low-level structure can lead to better likelihoods. See below for detailed feedback all of which are critical.

---

> ### Author Response · Authors · 2025-03-21
> **Response to reviewer zWos: Part 1**
>
> We thank the reviewer for their time and effort in reviewing our manuscript.  We will respond to each of the issues raised and outline the corresponding revisions made to the manuscript.  We plan to upload a fully revised manuscript by Monday 24 March 2025.
>
>
> > Comment:  In the introduction, the paper can add a bit more high-level motivation on studying CFM’s for out-of-distribution detection. For example, it is now known that normalizing flows do not provide reliable likelihoods. Why would the results/insights to be different as compared to discrete normalizing flows?
>
> **Response:**  Continuous normalising flows specify the transformation from the target distribution to the base distribution as the solution to an ordinary differential equation.  The inductive biases from this kind of transformation are different to those from discrete-step normalising flows, and the two do not necessarily behave in the same way.  We wanted to investigate whether continuous flows trained on image data also suffer from the phenomenon of unreliable likelihood, and whether a careful choice of feature representation can alleviate the problem.  We will emphasise this high-level motivation in the introduction.
>
>
> > Comment:  The authors hypothesize that absence of low-level image structure may lead to better OOD detection. It seems to me that this is more of correlation than causation. The paper may soften the claim with regards to this.
>
> **Response:**  A few of the reviewers raised this issue, and we agree.  We were not precise in our definition of “image-specific structure”, and our experimental evidence is insufficient to make the conclusion in question.  We decided to soften (and essentially remove) the claim that the absence of image-specific structure leads to more reliable likelihoods.  Instead, we will emphasise our focus on finding an autoencoder architecture and training scheme that leads to reliable likelihoods.  That some of the encoded features exhibit perceptual similarity to the input images is an interesting observation for which a more detailed investigation is reserved for future work.
>
>
> > Comment:  I assume that the features used are from the pooling layer of EffNet. The paper can train CFM Model on earlier feature representations above the pool layer which would retain more image-specific structure as compared to the pooling later. If the OOD detectability of these layers are worse than the pooling layer, this will add further evidence to this hypothesis. The final layer of EfficientNet may also lack low-level image structure but it performs worse on OOD as compared to PixelAE. So there are other factors that contribute to OOD detection. The authors can add some intuition on this in the paper.
>
> **Response:**  We appreciate these suggestions, and are keen to explore in greater depth the effect of “image-specific structures” (which in the updated manuscript we will refer to as perceptual similarity between the inputs and the encoded features).  We will reserve this for future work, and revise the current manuscript to emphasise our focus on whether training on feature representations can lead to reliable likelihoods.
>
>
> > Comment:  The authors train a baseline CFM model directly on pixels. However, the quality of this model is unclear to me so the authors may report the FID of this baseline. It does not have to be SOTA but it should be convincing that this model can generate high-quality images. The authors can similarly report FID on samples generated using the CFM on PercAE features and reconstructed with the decoder.
>
> **Response:** Our revised manuscript will include FID scores and generated samples from all the models (including the baselines and the CFM models trained on PercAE features).
>
>
> > Comment:  There have been at least a couple of works DDPM-OOD (https://arxiv.org/pdf/2211.07740) and LMD (https://arxiv.org/abs/2302.10326) that show diffusion models can reliably be used for out-of-detection. DDPM-OOD shows that this can be done with diffusion models trained in pixel space while this paper offers contradictory results. The paper can contextualize their work with these other techniques with some discussion. It is even better if they compare their approach with PixelAE features to DDPM-OOD on some of the datasets. For example, the authors can report the AUC score they obtain and compare it to other methods from Table 1 of DDPM-OOD.
>
> **Response:**  We thank the reviewer for pointing that out.  The related work section of our updated manuscript will include a discussion on those diffusion-based models.  They are typically trained through the optimisation of a lower bound on the log likelihood, and (unlike continuous flow models) do not provide exact likelihoods for data samples.  We therefore decided not to include them in our results section where we look specifically at likelihoods of test data.

---

> ### Author Response · Authors · 2025-03-21
> **Response to reviewer zWos: Part 2**
>
> > Comment:  Table 2 is also a bit unsatisfactory. For example, the LDA classifier can reliably predict only 75% of time that a predicted sample is from SVHN. This raises concerns on how good the CFM model on EfficientNet features is.
>
> **Response:**  Since generated ENet features cannot be decoded to images, we cannot compute FID scores for them.  We thought to run this experiment with the LDA classifier merely for a rough indication of whether these particular CFM models can produce samples close to their training distribution and far from the distributions underlying other datasets.  It is certainly not a rigorous test, but it is also not of central importance to the focus of our manuscript.  We will make this clear in the revised manuscript.
>
> > Comment:  More information on the PixelAE architecture is needed. For example: How does the architecture go from 2-D spatial representation to 1-D representation? If the bottleneck representation has no spatial information then how does the model reconstruct visually perfect images from this bottleneck? Are there skip connections from the downsampling blocks to the upsampling blocks?
>
> **Response:**  We will expand the description of the architecture.  There is, for example, a flattening layer between the last convolutional and the fully connected layers which converts the 2D feature maps to 1D.  The phrase “no spatial information” was misleading, and we will remove it.  As we mentioned above, we now view the presence or absence perceptual similarities between the inputs and the encoded features as an interesting observation, and will remove any claim that they are the cause of unreliable or reliable likelihoods.

---

### Review · Reviewer_f3k7 · 2025-01-07

**Summary Of Contributions:**

The authors tackle the problem of likelihood estimation in continuous-space conditional flow (CFMs), and show that the limitations of poor likelihood estimation, as observed in the discrete flow-matching setting, also translate to the continuous setting. Taking the example of MNIST, F-MNIST, CIFAR-10 and SVHN datasets, they propose to validate that the cause for this is the learning of the “feature-heavy” representation, be in thru pre-trained network, or auto encoders that preserve image features. Finally, they show that leveraging he latent space of an auto encoder trained on a pixel-level loss, not trained to model the image-space features, results in superior likelihood estimation, but at the cost of image quality.

**Audience:**

Yes

**Broader Impact Concerns:**

The Broader Impact section of the paper sufficiently addresses the ethical concerns.

**Claims And Evidence:**

No

**Requested Changes:**

Most of the questions have been laid out as part of the Weaknesses. For completeness, I’ll reiterate those that I feel are crucial to understanding/validating the key claims of the paper:

(1.a) Could the authors show a classifier trained on each of the latent spaces, just as in the EfficientNet case, to show that the same feature classes are no longer separable, which is why PixAE works best?

(1.c) It is still possible that the image-space features still cluster in the latent space on the PixAE. Would this invalidate the hypothesis?

(2.a) Is the limitation due to the loss — Could PixAE be trained with the same, lower, latent dimension, but the Perceptual loss?

(2.b) If PixAE output a (3136,1) dimensional vector, would reconstructions be better due to the clear over-parametrization of the latent space, just like in PercAE?

Answering the remaining questions, I believe, are for strengthening the claims, and giving a more holistic picture. Say, for example, an analysis of the latent space of higher-dimensional images, or ablating on other classifier models.

**Strengths And Weaknesses:**

**Strengths**:

 - While I am not an expert in the CFM space, I found that the paper presents the problem setting and the landscape of the literature comprehensively.
 - The proposed analysis is clearly described and well motivated.
 - The experimental setup and results are generally well explained.
 - The setting is relevant to the TMLR audience.

**Weaknesses**:

- **1. Validity of the Claim** I find that the proposed experimentations do not entirely support the claim that the *proposed results suggest that feature spaces preserving image-specific structure do not solve the problem of unreliable likelihoods*. If the intuition is that the perceptual autoencoder is the only one of the two AEs that retains/captures image-level feature information and therefore, models worse likelihood, then I would have expected a classifier to be trained on those latent spaces, just as in the EfficientNet case, to show that the same feature classes are no longer separable. (a) Could the authors show that this is/isn’t the case? The current approach is only to intuit based on the choice of loss, or rely on visual feedback. The latter seems to work for the chosen four datasets, but that could also be due to the fact that the latent and original dimensionalities, per-channel, are comparable. (b) Would this visual discernibility still hold when the perceptual AE is a down-projection, say when using the AE trained on ImageNet, CelebA-HQ, FFHQ, etc? (c) Alternatively, it is still possible that the image-space features still cluster in the latent space on the PixAE. Would this invalidate the hypothesis?
- **2. AE Ablations**: Is the reconstruction performance of the decoder a consequence of the architecture (including latent dimensionality) or the choice of perceptual loss function. While this is not important in general, given that we are trading out likelihood for generation capability (*i.e.,* the PixAE solution can no longer generate reliable images even on smaller datasets such as CIFAR-10 or SVHN), this seems like an analysis that needs to have been carried out. Could the PixAE model have been trained better to allow for reliable reconstructions, while still achieving good likelihood estimation. This leads to several question (a) Is the limitation due to the loss — Could PixAE be trained with the same, lower, latent dimension, but the Perceptual loss; (b) If PixAE output a (3136,1) dimensional vector, would reconstructions be better due to the clear over-parametrization of the latent space, just like in PercAE?
 - **3. Feature extractor ablations**: What is the training complexity of the CFM on the ENet representations. If this computational load is not high, the authors could have picked other encoders too, say CLIP, or more likely, the InceptionV3 embeddings. For example, the entire premise of the FID metric the authors talk about is that the Inception Embeddings are “nearly Gaussian” and could therefore be a reliable choice of latent embedding that is “feature agnostic.”
 - **4. Minor Nitpicks**: (a) The definition of the problem setting as an IVP is a bit confusing. Eqn. 1 defined the problem as solving $\frac{d z}{d t} = f(z,t)$ with $z_0 = u$ and $z_1=x$, which is a boundary value problem. In my understanding, the IVP would rather have defined $z_0 = u$ and $\frac{d z}{d t}|_{z=z_0} = u_1$. I found the cited literature to also not be consistent with this, so maybe I’m missing something here. (b) It would be good to include the PixAE loss explicitly in Sec 3.3., for a potential audience that is not familiar with auto encoders (c) Maybe I am unfamiliar with the literature of likelihood estimation for CFMs, but the log likelihood plots are unintuitive. It is not clear if (for example in Fig 1.a) it is bad if the OOD has “taller bins”, or “lies to the right” of the desired datasets’ log likelihood. The answer became clear to me only once I reached Fig.8. Adding some context for this plot, and also possible labels for the X and Y axis would help clear this up.

---

> ### Author Response · Authors · 2025-03-21
> **Response to reviewer f3k7**
>
> We thank the reviewer for their time and effort in reviewing our manuscript.  We will respond to each of the issues raised and outline the corresponding revisions made to the manuscript.  We plan to upload a fully revised manuscript by Monday 24 March 2025.
>
>
> > Comment:  I find that the proposed experimentations do not entirely support the claim that the proposed results suggest that feature spaces preserving image-specific structure do not solve the problem of unreliable likelihoods.  Could the authors show a classifier trained on each of the latent spaces, just as in the EfficientNet case, to show that the same feature classes are no longer separable, which is why PixAE works best? It is still possible that the image-space features still cluster in the latent space on the PixAE. Would this invalidate the hypothesis?
>
> **Response:** A few of the reviewers raised the issue in the first part of this comment, and we agree.  We will soften (and essentially remove) the claim that image-specific structure in the feature space do not solve the problem of unreliable likelihoods, and rather reserve a detailed investigation into the observed perceptual similarity between inputs and some of the feature representations for future work.  We do expect some class separation in the PercAE and PixAE feature spaces, since features can successfully be decoded into images from different datasets.  As such, LDA-type separability might not be the main cause for reliable or unreliable likelihoods.  Since ENet features cannot be decoded to images, we ran the experiment with the LDA classifier on ENet features merely for a rough indication of whether those CFM models can produce samples close to their training distribution and far from the distributions underlying other datasets.  This will be clarified in the revised manuscript.
>
>
> > Comment:  Is the limitation due to the loss — Could PixAE be trained with the same, lower, latent dimension, but the Perceptual loss?
>
> **Response:**  This is a valid question.  We will be including results from an autoencoder with the same architecture as PixAE, trained with the perceptual loss, in order to shed some light on the matter.
>
>
> > Comment:  If PixAE output a (3136,1) dimensional vector, would reconstructions be better due to the clear over-parametrization of the latent space, just like in PercAE?
>
> **Response:**  This is a good question.  We have not been able to investigate it fully within the time limitation, but would be keen to do so in future.  Our aim with this current work was to investigate whether continuous flows trained on image data also suffer from the phenomenon of unreliable likelihood, and whether a careful choice of feature representation can alleviate the problem.  We will emphasise this aim in the revised manuscript.
>
>
> > Comment:  Maybe I am unfamiliar with the literature of likelihood estimation for CFMs, but the log likelihood plots are unintuitive.  It is not clear if (for example in Fig 1.a) it is bad if the OOD has “taller bins”, or “lies to the right” of the desired datasets’ log likelihood.  The answer became clear to me only once I reached Fig.8.  Adding some context for this plot, and also possible labels for the X and Y axis would help clear this up.
>
> **Response:**  Histograms of log likelihoods over appropriate test sets are often used in the literature on flow-based generative models.  The histograms are normalised with respect to the number of samples per set, so that they can be interpreted as (empirical) distributions.  A taller and narrower histogram implies less variation in the likelihoods, a histogram more to the left implies lower likelihoods, etc.  We will add detail to give clearer intuition at the first mention of these kinds of plots in the manuscript.
>
>
> > Comment:  The definition of the problem setting as an IVP is a bit confusing. Eqn. 1 defined the problem as a  boundary value problem.
>
> **Response:**  We agree.  We do want to desribe an IVP, with the data the solution of the IVP at some time t_1.  We will remove the stated solution value at t_1 from Equation 1, and instead mention our interest in the solution at t_1 in the text below the definition.

---

### Review · Reviewer_dopc · 2025-03-09

**Summary Of Contributions:**

This paper studies the likelihood learned by CFM models under the features learned by different approaches, e.g., original data, classifier, PercAE and PixAE, and tries to compare their likelihood reliability against OOD data. The authors find that the CFM learnt by PixAE has more reliable likelihood and suggest the further direction to further improve the quality of generated samples of PixAE.

**Audience:**

Yes

**Broader Impact Concerns:**

The authors include the broader impact statement.

**Claims And Evidence:**

Yes

**Requested Changes:**

Please address the weaknesses above.

Additional comments:

1. What is signed Bhattacharya distances and FID for PercAE?

2. I suggest to combine the tables 3 and 4 to provide better comparison across all representations studied in the paper (figure 5. 6 and 8 as well)

3. Is it a trade-off between sample quality and likelihood reliability? It will be an interesting aspect to study. Like, when you improve the quality of PixAE, will its likelihood become slightly unreliable?

**Strengths And Weaknesses:**

Strengths:
1. The paper invests an interesting research questions on CFM model, as it could be potentially used in a critical application, like fraud detection, a reliable method is necessary.
2. The paper explored different representations widely used in the community and compared their pros and cons systematically.


Weaknesses:

1. As pointed out in the abstract, the authors provide a few application, like outlier detection, when the CFM model could not produce reliable likelihoods; however, there is no corresponding experiments to further support it, the current experimental setup indeed study the likelihood reliability from each data representation but how does it affect the downstream applications are not clear.

2. As an empirical study paper for machine learning tasks, the study for each feature category is limited; e.g., how about EfficientNet-B7 or transformer-based models? how about the model pretrained with larger dataset, like imagenet-21k?  how about the feature extracted by the model trained with masked auto-encoder? Moreover, is the observation consistent when moving to larger-scale datasets?

3. No holistic comparison among all feature representations, only one-to-one comparison at sec 4.1, 4.2, 4.3.

---

> ### Author Response · Authors · 2025-03-21
> **Response to reviewer dopc**
>
> We thank the reviewer for their time and effort in reviewing our manuscript.  We will respond to each of the issues raised and outline the corresponding revisions made to the manuscript.  We plan to upload a fully revised manuscript by Monday 24 March 2025.
>
> > Comment:  As pointed out in the abstract, the authors provide a few application, like outlier detection, when the CFM model could not produce reliable likelihoods; however, there is no corresponding experiments to further support it, the current experimental setup indeed study the likelihood reliability from each data representation but how does it affect the downstream applications are not clear.
>
> **Response:**  Our focus was on obtaining reliable likelihoods from continuous normalising flows.  Since these models do provide a mechanism for evaluating the likelihood of test samples, Improving the reliability of those likelihoods will directly influence the success of a downstream application like outlier detection (we can compare likelihoods of test samples to those of in-distribution samples, in order to ascertain whether the test samples are outliers).  We can make this clearer.
>
> > Comment:  As an empirical study paper for machine learning tasks, the study for each feature category is limited; e.g., how about EfficientNet-B7 or transformer-based models? how about the model pretrained with larger dataset, like imagenet-21k? how about the feature extracted by the model trained with masked auto-encoder? Moreover, is the observation consistent when moving to larger-scale datasets?
>
> **Response:**  We included features from EfficientNet-B4 specifically because of the study by Kirichenko et al. (2020) which did the same for discrete-step normalising flows.  We then abandoned features from pretrained classifiers (because they lack a decoder to convert generated feature vectors to images), and instead focused on autoencoders.  Our computational budget placed some restriction on dataset size, and we decided to first see if we can improve likelihood reliability on a smaller scale.
>
> > Comment:  No holistic comparison among all feature representations, only one-to-one comparison at sec 4.1, 4.2, 4.3. I suggest to combine the tables 3 and 4 to provide better comparison across all representations studied in the paper (figure 5. 6 and 8 as well). What is signed Bhattacharya distances and FID for PercAE?
>
> **Response:**  The reviewer makes a good point.  Our revised manuscript will include signed Bhattacharyya distances and FID scores for all the models (including the CFM models trained on PercAE features).
>
> > Comment:  Is it a trade-off between sample quality and likelihood reliability? It will be an interesting aspect to study. Like, when you improve the quality of PixAE, will its likelihood become slightly unreliable?
>
> **Response:**  This is an interesting question, and one we hope to explore in greater depth in future work.  In the revised manuscript, we will be including results from an autoencoder with the same architecture as PixAE, trained with the perceptual loss, in order to shed some light on the matter.

---

### Review · Reviewer_D9Em · 2025-03-10

**Summary Of Contributions:**

This manuscript studies the problem of distributional robustness in continuous flow models (CFMs). Specifically, their goal is to ensure that out-of-distribution data does not get assigned to out-of-distribution data.

The authors consider an approach where CFMs are trained with different featurization approaches (EfficientNet-B4 Features, and two auto-encoder latent feature approaches, one with a perceptual loss, and the other with the pixel-level loss). The autoencoder approaches are preferred, due to the presence of an inverting decoder.

The authors find that the approach pixel-level autoencoder is the best, significantly mitigating unreliable likelihoods on out-of distribution data.

**Audience:**

Yes

**Claims And Evidence:**

No

**Requested Changes:**

**Main Claim**

It seems like the main claim is that (image-specific structures in the data ==> unreliable likelihoods). If they want to pursue this claim, the authors should design a set of experiments that aim to validate this hypothesis. I do not think the current set of experiments does this.

1. Perhaps start by formalizing what "image-specific structure" means?

2. Explain how the Enet experiments might support/invalidate this hypothesis, as they presumably do not retain "image structure".

3. Then study some autoencoders that do/don't retain this structure

Alternatively, the authors should weaken their claim, and think more about meaningful conclusions we might be able to draw from these experiments.

**Other Points**

1. Re-organize the figures/tables so that better comparisons can be drawn.

2. More related work on relevant domain shift literature

3. See other comments in weakness section

**Strengths And Weaknesses:**

**Strengths**

1. I think the way the authors have presented the problem is very straightforward and approachable

2. The figures are nice, and really help with understanding.

**Weaknesses**

I find some aspects of the presentation to be rather *unscientific*. For example, the authors say in the abstract: "We show
empirically that representations containing image-specific structure still lead to unreliable likelihoods from CFM models.". I think this is a misrepresentation of what you have done. You have tested a *specific* perceptual autoencoder, and a specific pixel loss autoencoder, and insufficient evidence is provided to support this claim.

There are several other cases, where I found the authors claims to be unsubstantiated. I highlighted a few. Interspersing these kinds of comments in a scholarly articles is not appropriate. If you wish to make conjectures, consider adding a section for conjectures, and remove these kinds of statements.

> "We suspect that this might also be due to the fully convolutional nature of the autoencoder, as we found that a fully convolutional autoencoder trained with a pixel-based reconstruction loss also preserves image-specific structure in the feature space."

> "From this we suspect that CFM model likelihoods may depend on frequently occurring pixels, rather than semantic content."

*Note: To be clear, I'm not saying that I think the claim is false, it seem reasonable to me that this claim is true, due to correlations between representations across distributions imparted by the perceptual loss, I'm just saying, you have not provided convincing evidence of this claim*

Minor:

1. Since the figures are not available in one place, we can't quickly compare the various approaches.

2. Pg. 9 Table 3. over "multiple training runs?" Can you be more quantitative? Consider telling us the standard error of the number of training runs. Anyway, I think the standard error might be a better thing to report over the standard deviation, since I doubt most people would be able to interpret a variance in the Signed Bhattacharya distance.

3. Can you perhaps use a more standard way to evaluate the distance between distributions? KL, TV, etc.?

4. Wherever possible if you compute a evaluation, i.e., FID, compute it for all approaches.

---

> ### Author Response · Authors · 2025-03-21
> **Response to reviewer D9Em**
>
> We thank the reviewer for their time and effort in reviewing our manuscript.  We will respond to each of the issues raised and outline the corresponding revisions made to the manuscript.  We plan to upload a fully revised manuscript by Monday 24 March 2025.
>
> > Comment:  I find some aspects of the presentation to be rather unscientific. For example, the authors say in the abstract: "We show empirically that representations containing image-specific structure still lead to unreliable likelihoods from CFM models.". I think this is a misrepresentation of what you have done. You have tested a specific perceptual autoencoder, and a specific pixel loss autoencoder, and insufficient evidence is provided to support this claim.
>
> **Response:**  A few of the reviewers raised this issue, and we agree.  We were not precise in our definition of “image-specific structure”, and our experimental evidence is insufficient to make the conclusion in question.  We decided to soften (and essentially remove) the claim that the absence of image-specific structure leads to more reliable likelihoods.  Instead, we will emphasise our focus on finding an autoencoder architecture and training scheme that leads to reliable likelihoods.  That some of the encoded features exhibit perceptual similarity to the input images is an interesting observation for which a more detailed investigation is reserved for future work.
>
> > Comment:  There are several other cases, where I found the authors claims to be unsubstantiated. I highlighted a few. Interspersing these kinds of comments in a scholarly articles is not appropriate. If you wish to make conjectures, consider adding a section for conjectures, and remove these kinds of statements.
>
> **Response:**  We are removing these statements from the manuscript, and are taking more care in our interpretations of results.
>
> > Comment:  Re-organize the figures/tables so that better comparisons can be drawn.  Since the figures are not available in one place, we can't quickly compare the various approaches.  Wherever possible if you compute a evaluation, i.e., FID, compute it for all approaches.  Pg. 9 Table 3. over "multiple training runs?" Can you be more quantitative?
>
> **Response:**  We thank the reviewer for these suggestions.  Figures and tables will be re-organised as suggested, to ease comparison.  Our revised manuscript will also include signed Bhattacharyya distances, FID scores and generated samples from all the models.
>
> > Comment:  Consider telling us the standard error of the number of training runs. Anyway, I think the standard error might be a better thing to report over the standard deviation, since I doubt most people would be able to interpret a variance in the Signed Bhattacharya distance.  Can you perhaps use a more standard way to evaluate the distance between distributions? KL, TV, etc.?
>
> **Response:**  Reporting the mean and standard deviation over multiple training runs does seem to be slightly more common in related literature on flow-based models, but we will be more precise about the number of training runs so that the standard error can be calculated easily.  We could certainly have used a different way of measuring the difference between distributions, but chose the signed Bhattacharyya distance for its simplicity and close relation to the amount of overlap between two sets of samples.  Since the histograms we consider have relatively simple shapes, and we are interested mainly in how the distance might change for different datasets and different feature representations, we believe our conclusions would be largely unaffected by the specific metric used.

---

### Author Response · Authors · 2025-03-21
**General response to all reviewers**

We sincerely thank all the reviewers for their careful consideration of our manuscript, and their valuable comments.  We have taken the time to work through each review in detail, and tried to improve the manuscript accordingly.  We will respond to specific issues in separate replies to the individual reviewers.

Based on the reviewers’ suggestions, we decided to include additional results for the sake of completeness and more thorough comparisons.  We plan to upload a fully revised version of the manuscript (with the additional results and all the changes mentioned in our responses) by Monday 24 March 2025.

---

### Author Response · Authors · 2025-03-24
**Revised manuscript has been uploaded**

We have now uploaded a fully revised version of our manuscript that incorporates all the changes outlined in our responses. We have softened the claim regarding image-specific structure and included additional results to ensure completeness and provide a more thorough comparison.

---

### Decision · Action_Editor_H8Tg · 2025-05-12

**Recommendation:** Reject

**Comment:**

In its current form, there are some issues with the execution of the paper and supporting the claims. The reviewers unanimously recommended rejecting the paper in its current form, with the main reasons summarized above. Overall, I believe this paper would be of interest to the TMLR community, and I recommend that the authors revise the paper and submit a revision.

**Audience:**

OOD detection is an active area of research, and the paper would generally be of interest to the TMLR audience.

**Claims And Evidence:**

The paper studies out-of-distribution detection with conditional flow matching models (CFM), a continuous-time generalization of normalizing flows. The authors extend the results from prior work, showing that these models, similar to discrete-step normalizing flows, fail to detect OOD inputs when trained directly on pixels, but do better when trained on representations from another model. The authors study three different choices of representations: EfiicientNet and two autoencoders. They show that results differ depending on the choice of the encoding model.

The authors initially made claims suggesting that the success of OOD detections depends on the encoder removing "image-specific structure" from the representations. The reviewers argued that this claim was not supported by sufficient evidence, and it was removed in the updated manuscript. It was one of the core claims of the paper, stated many times in the text. It would be good to try to get more evidence for a version of this claim, rather than removing it.

The reviewers were also concerned with the quality of the models used, suggesting that the empirical results are overall poor. The authors should start with existing architectures for the encoder and CFM, so that it is possible to compare the results to the relevant baselines in the literature.

Generally, the study would be more complete if a wider range of encoder models were considered, including different model architectures and pretraining datasets and objective (supervised / autoencoder). It is also unclear if the results are at all specific to CFMs, or generalize to all types of likelihood models.

**Resubmission Of Major Revision:**

The authors may consider submitting a major revision at a later time.